

**Ozone and temperature decadal solar-cycle responses, and their relation to diurnal**
**variations in the stratosphere, mesosphere, and lower thermosphere, based on**
**measurements from SABER on TIMED.**
**Frank T. Huang[1*], Hans G. Mayr[2*]**
[1]University of Maryland, Baltimore County, MD 21250, USA
[2]NASA Goddard Space Flight Center, Greenbelt, MD 20771, USA
*retired
**Abstract.** There is evidence that the ozone and temperature responses to the solar cycle of ~11
years depend on the local times of measurements. Here we present relevant results based on
SABER data over a full diurnal cycle, not available previously. In this area, almost all satellite
data used are made at only one or two fixed local times, which can be different among various
satellites. Consequently, estimates of responses can be different depending on the specific data
set. Also, over years, due to orbital drift, the local times of measurements of some satellites have
also drifted. In contrast, SABER makes measurements at various local times, providing the
opportunity to estimate diurnal variations over 24 hrs. We can then also estimate responses to the
solar cycle over both a diurnal cycle and at the fixed local times of specific satellite data for
comparison. Our results of responses, based on zonal means of SABER measurements, agree
favorably with previous studies based on data from the HALOE instrument, which measured
data only at sunrise and sunset, thereby supporting the analysis of both studies. We find that for
ozone above ~ 40km, zonal means reflecting specific local times (e.g., 6, 12, 18, 24 hrs) lead to
different values of responses, and to different responses based on zonal means that are also
averages over the 24 hours of local time, as in 3D models. For temperature, effects of diurnal
variations on the responses are not negligible even at ~30 km and above. We also have
considered the consequences of local-time variations due to orbital drifts of certain operational
satellites, and for both ozone and temperature, their effects can be significant above ~30 km.
Previous studies based other satellite data do not describe their treatment, if any, of local times.
Some studies also analyzed data merged from different sources, with measurements made at
different local times. Generally, the results of these studies do not agree so well among
themselves. Although responses are a function of diurnal variations, this is not to say that they
are the major reason for the differences, as there are likely other data-related issues. The effects
due to satellite orbital drift may explain some unexpected variations in the responses, especially
above 40 km.
**1.0 Introduction**
The response of atmospheric ozone and temperature to the solar cycle of ~11 years is
important for both scientific and practical reasons. Global responses in the stratosphere,
mesosphere, and lower thermosphere have been investigated over decades based on a variety of
satellite data.
There is evidence that the values of the responses to decadal solar cycles depend on the local
times at which the measurements are made.
However, with few exceptions, the instruments on satellites measure at only one or two local
times, which are fixed for the entire mission.



Generally, previous studies do not address in detail the issue of diurnal variations of the
responses, and there have been no studies describing their variations over the 24 hrs of local
time. In the following, we provide estimates of the diurnal variations of the responses over a 24
hrs, which has not been available previously.
As noted in Huang et al. [2016b], previous global empirical results have been largely based
on data from the NOAA operational satellites (which include the Stratosphere Sounding Unit
(SSU), the Microwave Sounding Unit (MSU), and the Solar Backscatter Ultraviolet (SBUV)
instruments), from the Stratospheric Aerosol and Gas Experiment (SAGE I, II), on the Explorer
and Earth Radiation Budget (ERB) satellites, from the Halogen Occultation Experiment
(HALOE) on the Upper Atmosphere Research Satellite (UARS), and from the Sounding of the
Atmosphere using Broadband Emission Radiometry (SABER) instrument on the Thermosphere-
Ionosphere-Mesosphere-Energetics and Dynamics (TIMED) satellite, among others. The
advantage of the operational satellites is that they can provide global measurements covering
decades, being replaced as needed. However, issues of instrument offsets, stability, and
continuity over many years and decades can be problematic.
Except for SABER (and UARS), instruments on these satellites make measurements at only
one or two local times, which are fixed for the mission duration. The NOAA operational
satellites are sun-synchronous, in which case the measurements are made at two fixed local
times, one for the ascending orbital mode, and one for the descending mode. HALOE and SAGE
make solar occultation measurements, only at instrument sunrise and sunset. Consequently, used
as is, responses based on zonal means of the above measurements reflect long term variations at
the fixed local times, and could be a source of differences among the various studies.
They could also be a source of differences with 3D models, whose ozone amounts and
temperature vary with local time around a latitude circle, and whose zonal means are averages
over both longitude and 24 hrs of local time. When comparing results of responses based on
zonal means from measurements with models, Austin et al. [2008] point out that "The model
results are strictly zonal average values, which is an average over local time, whereas the
observations are typically made at fixed local times. Therefore, in the mesosphere, where the
diurnal variation of ozone is large, some of the differences between model results and
observations may have arisen from a diurnal variation in the actual solar response". See also
Beig et al. [2012].
In addition, the orbits of some operational satellites have drifted, so that the local times at
which the measurements are made have also drifted over several hours or more (see McPeters et
al. [2013], Frith et al. [2014], Remsberg [2008], Randel et al. [2009], Tummon et al. [2015],
Hood et al. [2015]). Tumman et al. [2015] summarizes some of the data processing methods
taken by various groups. Generally, they report that diurnal variations are either neglected, or are
assumed to be negligible below ~ 45-50 km. See also Davis et al. (2015).
Previous results have not generally agreed so well with one another in their details. A major
reason for these differences may be the conditions and constraints under which the various
measurements were made (Austin et al., 2008, Crooks and Gray [2005], Gray et al. [2005],
Huang et al. [2016b]).
In addition, previous studies generally have not described how they treat diurnal variations, so
that comparisons related to responses as a function of local times are problematical. We are also
not aware of studies based on orbital drift.
In contrast to most other measurements, SABER provide additional information which allows
us to estimate daily ozone and temperature diurnal variations, and then also the dependence of





their responses to the decadal solar cycle on local time. In the following, we focus on zonal
means of ozone and temperature, either at various specific local times, or averaged over local
times (as in 3D model), and the effects of their diurnal variations on their responses to solar
variability over a solar cycle of ~11 years (2002-2014), from 20 to 100 km.
97       In this study, we find that not only do the values of the responses depend on the local times at
which the measurements are made, but they can be significant even at altitudes as low as 30 km.
99       In Section 2, we review our previous analysis and derivation of diurnal variations and zonal
means that are averages of both longitude and local time around a latitude circle, based on
SABER measurements. We also describe how we can estimate new results of zonal means
corresponding to specific local times, and new results in estimating effects of orbital drift on
diurnal variations.
In Section 3 we describe our new results of responses to the solar cycle at the specific local
times of sunrise (6hrs) and sunset (18hrs), and compare with results from HALOE. This gives an
indication of the quality and reality of our and HALOE's results.
In Section 4 we describe our new results of responses to the solar cycle over a diurnal cycle of
24 hrs.
In Section 5 we describe our estimates of responses in situations where the local times have
'drifted' due to satellite orbital drifts. We also describe some previous studies.
In Section 6 we discuss the issue of data length.
**2.0 SABER analysis.**
The data are provided by the SABER project (version 2.0, level2A). They are interpolated to 4-
degree latitude and 2.5 km altitude grids, after which zonal averages are taken for analysis.
In contrast to other satellite measurements, those from SABER (Russell et al., 1999) contain
information to estimate the diurnal variations of ozone and temperature, and the results are
described in Huang et al. [2010a, 2010b].
As noted in Huang et al. [2016b], SABER ozone and temperature measurements have been
analyzed with success for more than a decade. We have derived variations with periods from one
day or less (diurnal variations) up to multiple years (semiannual oscillations (SAO) and quasi-
biennial oscillations (QBO)), and one decade or more (trends, responses to solar cycle). See
Huang et al. [2008a,b, 2010a,b, 2014, 2016a,b]. Zhang et al. [2006] and Mukhtarov et al. [2009]
have derived temperature diurnal tides using SABER data, and Nath and Sridharan [2014] have
also derived responses to solar variability using SABER data.
For both ozone and temperature, these studies show that, for variations that are deviations from
a mean state (e.g., diurnal variations, tides, semiannual and quasi-biennial oscillations, responses
to solar variability, trends), SABER measurements are robust and precise. For example, zonal
mean tidal temperatures can agree with other measurements to within ~ 1°K (Huang et al.,
2010a), and our zonal mean ozone diurnal variations can agree with other diurnal measurements
to less than a few percent (Huang et al., 2010b).
These previous results contain
1) diurnal variations of ozone and temperature for each day of the year, and
2) zonal means that are averages over both longitude and local time in a consistent manner,
which can then be compared directly with 3D models.
Using these, we can then estimate the goals of this study, which is to
3)  reconstruct the zonal means to reflect specific local times.


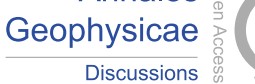

4) calculate responses to solar variability over a solar cycle at specific local times
5)  estimate local time variations of responses as a result of orbital drifts of NOAA satellites,
as noted above.
We can therefore find the variation of responses to the solar cycle over the 24hrs of local time,
including at 6 and 18hrs for comparison with responses based on HALOE data at sunrise and
sunset for comparison (see Beig et al. [2012], Fadnavis and Beig [2006])..
Compared to the stratosphere, diurnal variations of ozone and temperature themselves are
more prominent in the mesosphere and lower thermosphere. Even in the stratosphere, they may
not be negligible (Huang et al. 2010a, 2010b). Between ~30 and 80 km, ozone diurnal variations
are due mainly to photochemistry (Brasseur and Solomon, 2005), while temperature diurnal
variations are mainly a result of thermal tides (Chapman and Lindzen, 1970). For diurnal
variations, our results for both ozone and temperature (Huang et al. 2010a, 2010b) show that they
can be systematic from the lower thermosphere down to 25 km. This is consistent with results by
Sakazaki et al. [2015] for ozone, and Oberheide et al.[2000] and Gille et al. [1991] for
temperature.
As discussed below, for responses to the solar cycle, our results show that the effects of local
time variations can be non-negligible for altitudes even below 40 km, especially for temperature.
**2.1 Previous analysis**
**2.1.1 Diurnal variations**
As noted in Huang et al. [2016b], unlike other satellites mentioned above (except UARS), the
orbital characteristics of TIMED are such that SABER samples over the 24 hrs of local time,
which can be used to estimate diurnal variations of ozone and temperature. A complication is
that it takes SABER 60 days to sample over the 24 hrs of local time. Over 60 days, the variations
with local time are embedded with the seasonal variations, and need to be separated from them.
The method we use estimates both the diurnal and mean variations (e.g., seasonal, semiannual,
annual) together, by performing a least squares fit of a two-dimensional Fourier series, where the
independent variables are local time and day of year. The algorithm is discussed further in Huang
et al. [2010a,b].
The top row of Figure 1 shows zonal mean ozone diurnal variations (percent deviation from
midnight) for day 85 of 2005, at the equator, from 25 to 40 km (left panel), 45 to 60 km (right
panel), based on SABER data. See Huang et al. [2010b] for details. It can be seen that diurnal
variations can be significant even at 25 km.
The bottom row of Figure 1 corresponds to the top row, but for temperature. See Huang et al.
[2010a] for details. Even at altitudes near 30 km, the diurnal variations are systematic and, as
seen below, can affect results in estimating decadal responses.  Although small, at 30 km, the
diurnal variations of temperature compare well with Zeng et al. [2008], Oberheide et al.[2000],
Gille et al.[1991], based on different types of measurements.





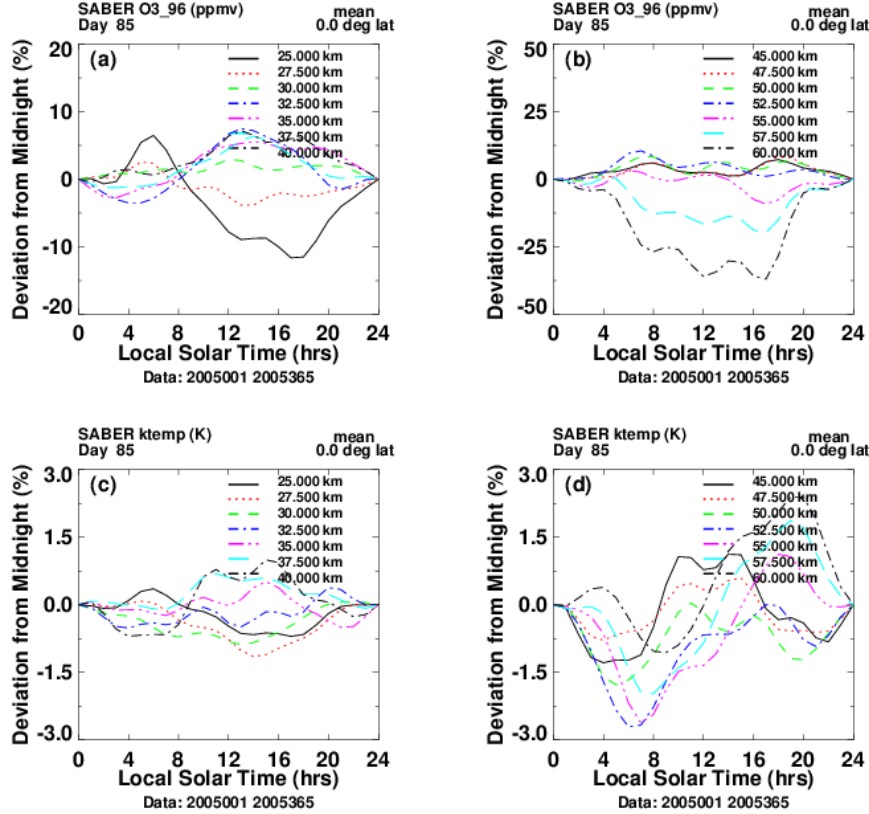

**Figure 1.** Top row: ozone zonal mean mixing ratios (ppmv) versus local time for day 05085 at the Equator. Left panel (a): 25 to 40 km (percent deviation from midnight), right panel (b): 45 to 60 km. Bottom row: as in top row, but for temperature (K).

**2.1.2 Mean variations.**

Once the diurnal variations are known for each day, the zonal mean variations, which are averages over longitude and local time, consistent with 3D models, can be obtained.

Based on these zonal means, our results of decadal responses to solar activity, as represented by the 10.7 cm solar flux, had been presented in Huang et al. [2016a, 2016b].

**2.2 Current analysis**

For the current study, we generate

a) monthly zonal means that are averaged over longitude, but at specific local times. These correspond to those satellite measurements which sample at specific local times

b) zonal means with local times that vary from month to month, to simulate the situation caused by satellite orbital drifts, as described earlier.

c) estimates of responses to solar the cycle, based on a) and b), and compare with responses based on zonal means that are also averaged over local time.

As an example, in Figure 2, the left panel (a) shows our ozone monthly mean mixing ratios (red line, parts per million by volume, ppmv) at 47.5 km and the Equator, from mid 2002 to mid




2014, with seasonal and local time variations removed. The green line represents how the data
would vary if we simulated the variations with local time due to orbital drifts of the NOAA
operational satellites. We have varied the local times such that from 2002 to 2014, they progress
from 12 to 18 hrs. Also shown is the corresponding 10.7 cm flux (black lines, right axis, units in
sfu). As can be seen, year 2002 was near solar maximum; the middle of solar cycle 23, and 2014
is some years into cycle 24, which began ~2008. The right panel (b) corresponds to the left
panel, but for temperature (K) at 45 km. The labels 'CRC' denote the correlation coefficients
between the respective ozone and temperature zonal means and the 10.7 cm flux.
The estimates of responses to the solar cycle are made using Equation (1), in a similar manner
as previously done by others, and by us, using a multiple regression analysis (e.g., Keckut et al.
[2005], Soukharev and Hood [2006], Huang et al. [2016b]) that includes solar activity, trends,
seasonal, quasi biennial oscillations (QBO), and local time terms, among others, on monthly
values. Specifically, the estimates are found from the equation

$$M(t) = a + b*t + d*F107(t) + c*S(t) + l*lst(t) + g*QBO(t) \qquad (1)$$

where t is time (months), a is a constant, b is the trend , $d$ the coefficient for solar activity (10.7
cm flux), c is the coefficient for the seasonal *(S(t))* variations,  $l$ the coefficient for local time *(lst)*
variations, and $g$ the coefficient for the QBO. As is often done, the seasonal and local time
variations are removed first, but we include them in Equation (1) for completeness. The F107
stands for the solar 10.7 cm flux, which is commonly used as a measure of solar activity, and the
values used here are monthly means provided by NOAA.
M(t) stands for the input ozone or temperature zonal means described in a) and b), above.
The algorithm is applied to the monthly zonal-mean values from June 2002 through June 2014
(as in Figure 2), from 48°S to 48°N latitude, and from 20 to 100 km.

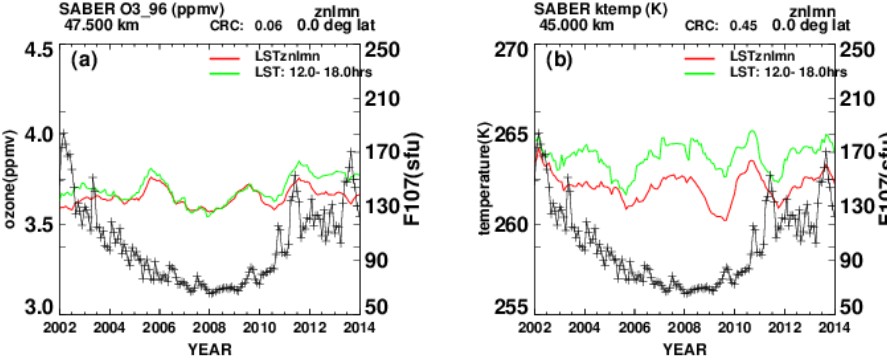

**Figure 2.** Ozone zonal mean mixing ratios (left panel, red line, ppmv) from mid 2002 to mid 2014, 47.5 km, 0º lat;
right panel, as in left panel, but for temperature (K) at 45km. Black lines (+, right scale) show the corresponding
monthly 10.7 cm flux (sfu) provided by NOAA.

**3.0 Results: Ozone and temperature responses to solar cycle at 6, 18hrs (sunrise and sunset)**
Specifically, we use the term 'response to solar activity (solar cycle)' generally to refer to the
term d*F107 in Equation (1), and in particular to ozone or temperature responses at solar
maximum minus those at solar minimum, per 100 solar flux units (sfu). For ozone, it is also in

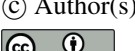

terms of percentage differences. A positive response means that the response at solar maximum
is larger than that at solar minimum (Huang et al.,2016b).

For the new results of this study, we focus on the following:
1) Responses to the solar cycle at 6 and 18 hrs (sunrise, sunset). Comparisons with
responses based on HALOE data (Beig et al. [2012], Fadnavis and Beig [2006]), which measure
only at sunrise and sunset.
2) Responses based on zonal means at specific local times.
3) Responses with local times changing due to satellite orbital drifts.
4) Comparison with results based on zonal means that are averages over both longitude and
local time simultaneously, as in 3D models.

**3.1 Ozone responses at 6, 18hrs (sunrise and sunset)**
We consider first sunrise and sunset (6, 18hrs) because there are direct empirical results with
which to compare, by Beig et al., [2012] and Fadnavis and Beig [20006], based on HALOE data
from January 1992 to November 2005. Importantly, unlike other studies, they describe how they
treat variations with local times, although they have results only at 6 and 18hrs.
The comparisons will indicate the quality of our results at 6 and 8hrs, and also over the 24 hrs
of local time.
In Figure 3 and applicable other figures, we have manually transferred values of plots from
other studies for comparison, so they are not exact, but should be adequate for our purposes.
In comparisons with results based on HALOE data, uncertainties should be considered.
According to Beig et al., [2012] and Fadnavis and Beig [20006], due to the sparse sampling
inherent in solar occultation measurements, there are only 8 to 12 data points (sometimes less)
per month for each latitude. So they generally present responses that are based on data
composited over 30-degree latitude bins (e.g., 0-30ºS, N) and averages of responses at sunrise
and sunset. We get results at 4-degree intervals. Even if we composite the SABER data into 30º
bins, the distribution within the bins would be uniform, but quite different than that of HALOE
data, so we will present our results at specific latitudes. Our responses can vary significantly as a
function of latitude, so that is another consideration in the comparisons.
In addition, here and in the literature, ozone responses are normally given in terms of percent
changes, and the value of the ozone itself is needed to get percent values. Because absolute
values among various instruments can sometimes be offset, it is an added source of uncertainty.
Figure 3 (left panel) shows our and that of Beig et al.,[2012] ozone responses from 50 to 100
km, at 4ºN. The magenta triangles show responses based on HALOE data for ozone (composite,
0-30ºN, BEIGN), which are averages of sunrise and sunset responses, and should be compared
with the red plusses, which denote the average of our results at 6hrs and 18hrs. It can be seen that
the agreement of our averages (magenta triangles and red plusses) are very favorable, except for
our large negative value at 77.5 km, and above 90km. As shown in Figure 4 (left panel), the
results of Beig et al., 2012] for 6hrs and 0º also show a large negative value near 75 km. It is
their values at 18 hrs (right panel) that seem anomalous (aside from what is shown in Figure 4,
Beig et al.,[2012] do not provide results separately for 6 and 18hrs). The green asterisks denote
our results for 6hrs and the blue diamonds denote our responses at 18 hrs. The right panel
corresponds to the left panel, but for 20ºN and 20 to 60km, and the HALOE results are from
Fadnavis and Beig [2006], 0-30ºN composite. As in the left panel, the agreements of our
averages (magenta triangles and red plusses) are very favorable. It can be seen that even in the
stratosphere, the responses at 6hr are different from those at 18hrs.




Considering our discussion of uncertainties above, we believe that the results of Beig et al.
[2012] and Fadnavis and Beig [2006] (magenta triangles), agree very well with our estimates
(red plusses) in both altitude ranges (both panels of Figure 3). Note in particular the rapid change
from negative to positive values near 75-80 km. In Figure 3, the left panel at 4ºN was chosen in
part to compare further with Figure 4, and the right panel at 20ºN was chosen to compare with
Beig et al.,[2012] results based on composite data in the 0-30º latitude band. We note that our
results show that there can be significant differences of responses at various latitudes.

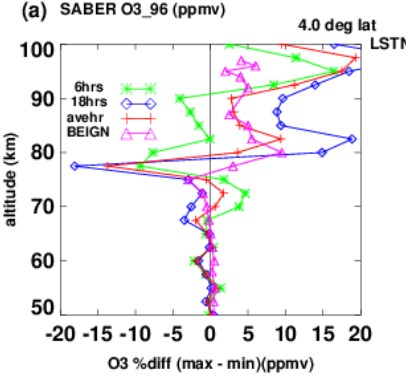
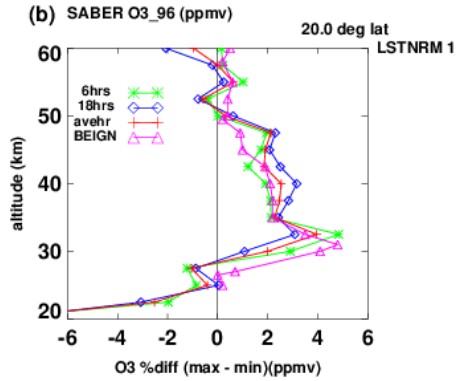

**Figure 3.** Ozone responses to solar activity versus altitude, at 4ºN, from 50 to 100 km (left panel), and 20ºN, from
20 to 60km (right). Values are responses at solar max minus responses at solar min (% /100sfu). Magenta triangles
denote results by Beig et al. [2012], average of responses at 6 and 18 hrs local time, and  0-30ºN.  Red plusses
denote our estimate (average at 6 and 18 hrs). Green asterisks denote our estimate at 6hrs, and blue diamonds,
estimate at 18hrs.

Figure 4 shows ozone responses to solar activity versus altitude, from 50 to 100 km, at the
equator for sunrise (left) and sunset (right). Values are responses at solar max minus those at
solar min (% /100sfu).  Red diamonds denote responses found by Beig et al. [2012] at 6 hrs (left
panel) and 18 hrs (right), composite from 0-4ºN. Blue plusses denote our corresponding results
based on SABER data.
It is the only instance where Beig et al.,[2012] show responses separately for 6 and 18hrs.
Except for the large negative values (red diamonds) from Beig et al [2012] in the left panel
near 74 km, and the large negative value (blue plusses) by us at 77.5 km in the right panel, we
believe that the comparisons are mostly favorable, in view of uncertainties discussed earlier.
Although not shown, the half width of the error bars provided by Beig et al.,[2012] between 80
to 90 km are ~± 10 ((% /100sfu)
This can be compared with our results in the left panel of Figure 3 at 4ºN. It is seen that
although there are sharp variations above 70km, the agreements are at least qualitatively good,
considering the caveats noted above.
The large excursions near 75 km are not isolated, but are systematic for both Beig et al.,
[2012] and us, as can be seen further in Figure 6 for 16ºN.



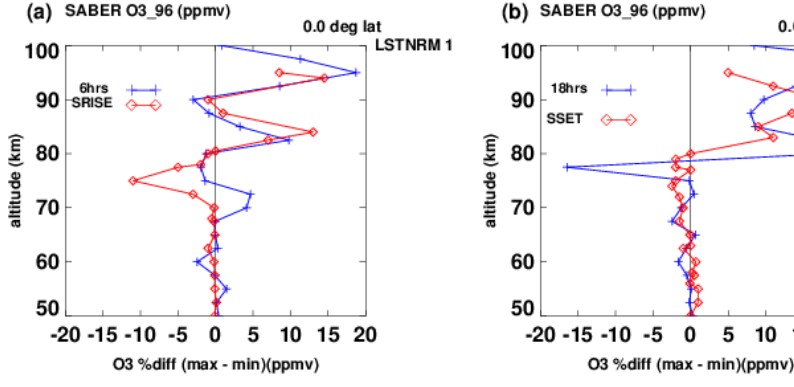

**Figure 4.** Ozone responses to solar activity versus altitude, from 50 to 100 km, at the equator. Values are responses
at solar max minus responses at solar min (% /100sfu). Left panel: Red diamonds denote results by Beig et al.
[2012] at 6 hrs (left panel) and 18 hrs (right) local time, composite from 0-4ºN. Blue plusses denote our results based
on SABER data at 6hrs and 0 deg (left panel) and 18hrs (right).

**3.2 Results: Temperature responses at 6, 18hrs (sunrise and sunset)**
Figure 5 corresponds to Figure 3, but for temperature. Values are responses at solar max minus
responses at solar min (ºK /100sfu).
The left panel shows our and Beig et al.,[2012] temperature responses from 50 to 100 km, at
32ºN. The magenta triangles show responses based on HALOE data, by Beig et al. [2012] for
temperature (composite, 0-30ºN, BEIGN), which are averages of sunrise and sunset responses,
and should be compared with the red plusses which denote the average of our results at 6hrs and
18hrs. It can be seen that the agreement of our averages (magenta triangles and red plusses) are
very favorable, except at 75km. Beig et al.,[2012] do not provide temperature responses above
75 km. The green asterisks denote our results for 6hrs and the blue diamonds denote our
responses at 18 hrs. Beig et al,.[2012] do not provide results separately for 6 and 18hrs.
The right panel corresponds to the left panel, but at16ºN and 20 to 60km, and the HALOE
results are from Fadnavis and Beig [2006], 0-30ºN composite. Above 30km, the agreements of
our averages (magenta triangles and red plusses) are very favorable. We note that according to
Fadnivas and Beig [2006] and Remsberg et al. [2002], that at altitudes below ~35km (~5hPa),
HALOE uses temperatures from the National Center for Environmental Prediction (NCEP).
This could be the reason for the differences between the magenta triangles and our red plusses
below 35 km.
It can be seen that even in the stratosphere, the responses at 6hr are different from those at
18hrs. We note that the left panel represents results at 32ºN, instead of 16ºN, as the agreement
with results by Beig et al. [2012] is somewhat better.





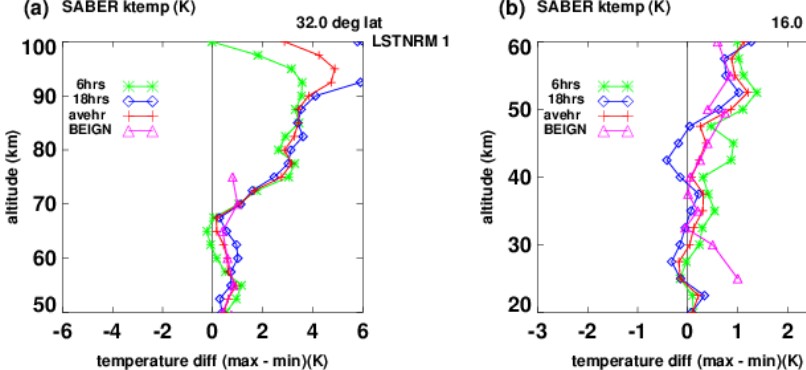

**Figure 5.** Corresponds to Figure 3, but for temperature responses to solar activity versus altitude, from 50 to 100 km (left panel), and 20 to 60 km (right). Values are responses at solar max minus responses at solar min ºK /100sfu. Magenta triangles denote results by Beig et al. [2012], averaged of 6 and 18 hrs local time (composite 0-30ºN). Red plusses denote our estimate (average of 6 and 18 hrs, at 32ºN (left panel)) and 16ºN, right panel), based on SABER data. Green asterisks denote our estimates at 6hrs, and blue diamonds are estimates at 18hrs.

**4.0 Ozone and temperature responses over a diurnal cycle.**

In this section, we extend our results to other local times.

Generally, previous studies based on other satellite measurements do not describe how they treat data with respect to local times, and we cannot make comparisons as with HALOE. Some studies use different data from various instruments, which mix data measured at different local times. See Section 5.2 and the discussion in reference to Figure 9, for details.

Figure 6 shows our ozone (left panel) and temperature (right panel) responses from 50 to 100 km, at 16ºN over a diurnal cycle (6, 12, 18, 24hrs). The black line denotes our responses based on SABER data where the zonal means are averages over both longitude and 24 hrs of local time. The green asterisks denote responses for 6hrs, blue diamonds (12hrs), red plusses (18hrs), and magenta triangles (24 hrs).

Up to this point, ozone values are responses at solar max minus responses at solar min (percent/100sfu). In the following, note that unlike the situation above at 6 and 18hrs for ozone at specific local times, the normalizing values used to obtain responses in percent are now averaged over local time, to be consistent with responses based on zonal means that are averages over both longitude and local time (black line in Figure 6).



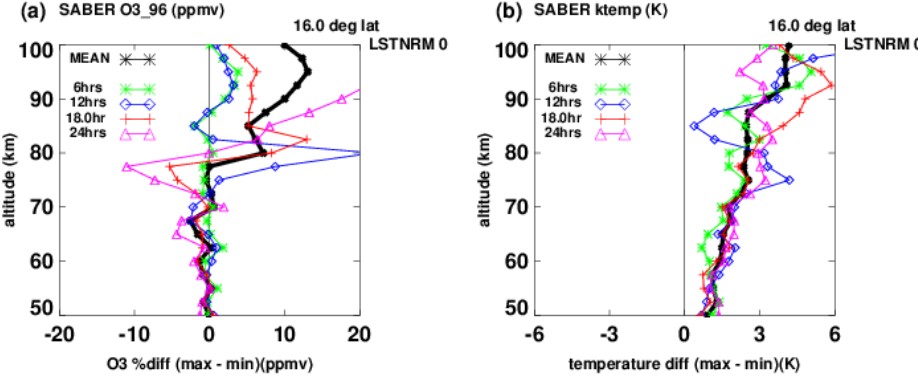

**Figure 6.** Ozone (left panel) and temperature (right) responses from 50 to 100 km at 16ºN. Values are responses
at solar max minus responses at solar min (% /100sfu) for ozone and ºK/100sfu for temperature. Black asterisks
denote responses based on zonal means that are averages over both longitude and local time. Green asterisks denote
our responses based on zonal means fixed at 6hrs, blue diamonds fixed at 12hrs, red plusses at 18 hrs, and magenta
triangles at 24hr, based on SABER data.
Figure 7 shows the ozone (left panel) and temperature (right panel) responses to solar activity
versus altitude, at the Equator, from 20 to 60 km, at 6hrs (green asterisks), 12hrs (blue
diamonds), 18hrs (red plusses), 24 hrs (magenta triangles), and based on zonal means that are
averages over local times (black asterisks). For ozone, below about 40 km, diurnal variations
have relatively little effect on responses. For temperature, the effects can be larger, even at
altitudes as low as 30 km.

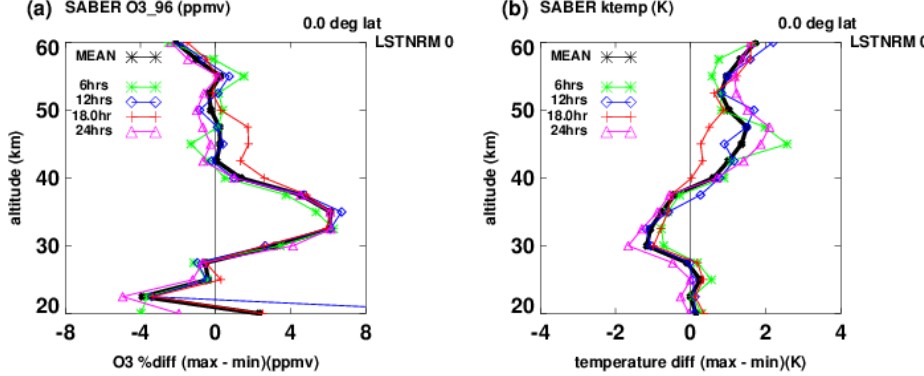

**Figure 7.** As in Figure 6, but from 20 to 60 km. Ozone (left panel) and temperature (right) responses at 0º. Values are
responses at solar max minus responses at solar min (% /100sfu) for ozone and ºK/100sfu for temperature. Black
asterisks denote our responses based on zonal means that are averages over both longitude and local time. Green
asterisks denote our responses of zonal means at 6hrs, blue diamonds at 12hrs, red plusses at 18 hrs, and magenta
triangles at 24hrs, based on SABER data.
**5.0 Comparisons with responses based on operational satellite measurements (fixed or**
**drifting local times).**




In the stratosphere and lower mesosphere, previous global results have been largely based on
data from the NOAA operational satellites (including the Stratosphere Sounding Unit (SSU), the
Microwave Sounding Unit (MSU), and the Solar Backscatter Ultraviolet (SBUV) instruments).
An advantage of the operational satellites is that they can provide global measurements covering
decades, being replaced as the instruments degrade. However, issues of calibration, instrument
offsets, stability, and continuity, can be problematical. The satellites are generally polar orbiters
and sun-synchronous, and make measurements at two fixed local times, one for the satellite
ascending mode, and one for the descending mode.
As noted above, in merging data from different satellites, consistency in local times needs to
be considered. Also, in some cases, over years, the orbits have drifted, so that the local times at
which measurements are made have also drifted by several hours. See McPeters et al. [2013],
Frith et al. [2014], Tummon et al. [2015], Hood et al., [2015]. Tumman et al. [2015] summarizes
some of the data processing methods taken by various groups. Generally, they report that diurnal
variations are either neglected, or are assumed to be negligible below ~ 45-50 km. See also Davis
et al. (2015).
Unlike Beig et al.,[2012], the various studies generally did  not address the issue of diurnal
variations in detail.
**5.1 Effects of local time variations due to satellite orbital drift**
To study the effects of local time changes due to orbital drift, from our estimates of diurnal
variations, we can simulate their effects on responses to solar variability. As a simple example,
Figure 8 shows our results for ozone (left panel) and temperature (right panel) responses to solar
activity versus altitude, at the Equator, from 20 to 60 km. Values are responses at solar max
minus responses at solar min in percent/100 sfu for ozone, and K/100 sfu for temperature. The
red squares denote results where local times increased linearly from 12 to 18 hrs from 2002 to
2014, to simulate orbital drift. Black asterisks denote responses based on zonal means that are
averages over both longitude and local time. It can be seen that there are significant differences
between them, especially above 40 km. We have also run tests with the local time varying at
different hours and durations, and the differences can be smaller or more pronounced than that
shown in Figure 8.

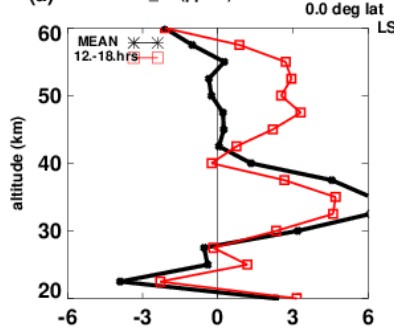
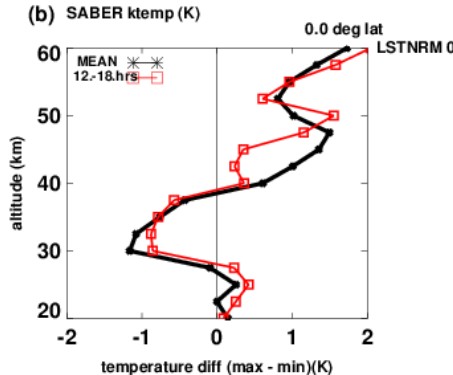

**Figure 8.** Ozone (left panel) and temperature (right panel) responses to solar activity versus altitude, at the Equator,
from 20 to 60 km. Values are responses at solar max minus responses at solar min in % per 100 sfu for ozone, and
K/100 sfu for temperature. Black asterisks denote responses based on zonal means that are averages over both

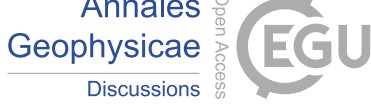



longitude and local time. Red squares denote corresponding results, but with local times increasing linearly from 12
to 18 hrs from 2002 to 2014.
**5.2 Comparisons with operational satellite data**
Figure 9 is taken from our previous analysis (Huang et al. [2016b], Figure 3). It compares
results from previous studies done by others, which were manually transferred by us, so they are
not exact. Our ozone responses (black line, SABER) are shown in the left plot (a), versus altitude
from 20 to 60 km, averaged from 24°S to 24°N, to better conform to results by others. The light
blue squares represent results of Remsberg (2008, RMSBRG), the green asterisks are from
Fadnavis and Beig (2006, BEIGN, 0-30°N), and the blue diamonds are from Beig et al.,(2012,
BEIGS, 0-30°S), all based on HALOE data.
The red line (plusses) in Figure 9(a) show ozone responses from Soukharev and Hood [2006]
(AUDTA, data from1979-2003), as reported by Austin et al. [2008], and from models (AUMDL,
magenta lines and triangles), also reported by Austin et al. [2008], representing composite results
from 25ºS to 25ºN latitude. The Soukharev and Hood [2006] results (red plusses) are a
composite based on SBUV, HALOE, and SAGE data, that show a minimum near 30 km, and a
maximum above 40 km.
The right plot in Figure 9(b) corresponds to the left plot, but for temperature. The temperature
responses (AUDTA, data from 1979-1997) were taken by Austin et al. [2008] from Scaife et al.
[2000]. In Figure 9(b), the black line denotes our responses based on SABER data, averaged
from 24°S to 24°N, to conform to previous results by others.
The issue of local time effects is not discussed in detail in these studies. As noted above,
Austin et al.,[2008] note that zonal means of models are averages over local time in contrast to
those based on satellite measurements, which are typically at fixed local times.

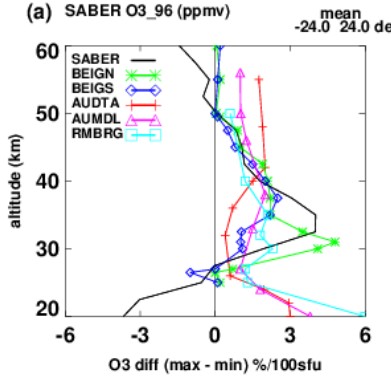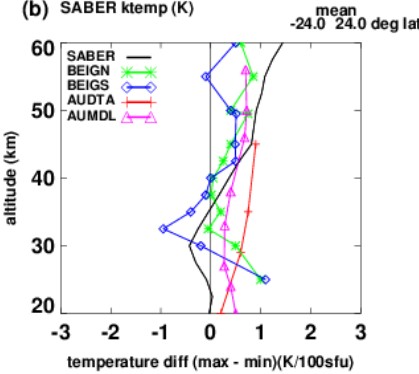

**Figure 9.** Left panel (a): ozone responses versus altitude from 20 to 60 km;  black line: SABER results averaged
from 24°S to 24°N; light blue squares: Remsberg (2008, RMSBRG); green asterisks: Fadnavis and Beig, [2006],
BEIGN, 0-30°N; blue diamonds :BEIGS, 0-30°S, HALOE data; red plusses: Austin et al. [2008]  data AUDTA;
magenta triangles, Austin et al., [2008] model, AUMDL, 25ºS to 25ºN latitude composite. Right panel (b):
temperature responses corresponding to left panel.
Nath and Sridharan [2014] have also analyzed the same SABER data as we did and derived
responses at 10–15º latitude. Plots comparing with our results are given in Figure 10 (taken from
Figure 5 of Huang et al. [2016a]). Black lines denote our results and red asterisks denote that by
Nath and Sridharan [2014]. For both ozone and temperature, their responses agree better with





ours up to ~45km, but not so well at higher altitudes. We believe that the differences of the
responses at higher altitudes are due to the local time variations in the SABER data, as discussed
in Section 2. Nath and Sridharan (2014) do not appear to have considered diurnal variations.
Note that in Figure 10 the ozone responses are not in percent differences, as in other plots, so that
differences between 45 and 80 km are not readily discernible, due to their small values.

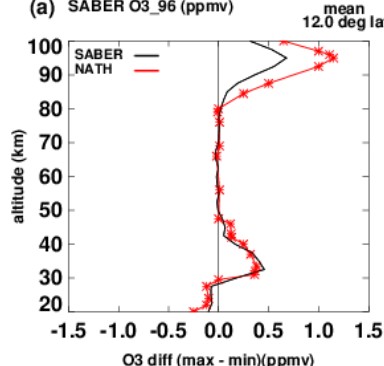
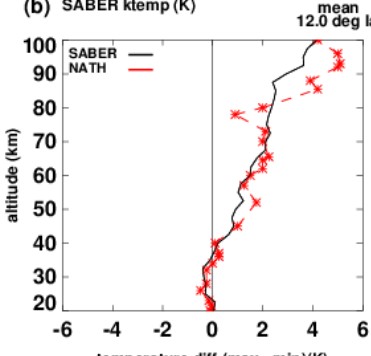

**Figure 10.** Ozone (left) and temperature (right) responses to solar activity vs. altitude, from 20 to 100 km. Values
are responses at solar max minus responses at solar min in ppmv /100 sfu for ozone and K/100 sfu) for temperature.
Black lines denote SABER responses at 12° lat; red color denotes results of Nath and Sridharan (2014), for 10–15°
lat, also based on SABER data.

**6.0 Time span of measurements.**
Figure 11 is a scatter diagram plot of monthly values versus the 10.7 cm flux. The top row
shows ozone at 47.5 km and the Equator, the bottom row shows temperature at 45 km and the
Equator. The left panels represent the monthly zonal means that are averaged over both longitude
and local time, and the right panels use zonal means where the local times simulate orbital drift
as discussed in reference to Figure 8. The red lines in Figure 11 represent linear fits between the
monthly values and the 10.7 cm flux, which corresponds to using only the solar term of the
multiple regression (Eq. 1). For ozone (top row), the values 0.28 percent/100sfu (left header
label, left panel) and 3.24 percent/100sfu at 47.5 km (right panel) compare well with the
regression results which uses all terms of Eq. (1), seen in Figure 8 (left panel). For temperature
(bottom row), the values 1.23K/100sfu and 0.35K/100sfu at 45 km also compare well with the
right panel of Figure 8. Consequently, aliasing from other terms in Equation (1) is not
significant. Unlike time series data, where time increases monotonically with data length, the
10.7 cm flux values remain within a fixed interval between solar minimum and solar maximum
(~70 and 200 sfu). In Fig. 11, the values span about one solar cycle. But even over more solar
cycles, the 10.7 cm flux values would only repeat and backfill in with values in the same general
area in Figure 11, effectively providing a more average result but not necessarily reducing the
uncertainty much otherwise.
It can be argued that even with more than one solar cycle of data available, analysis over
individual cycles should be made to analyze differences among solar cycles.





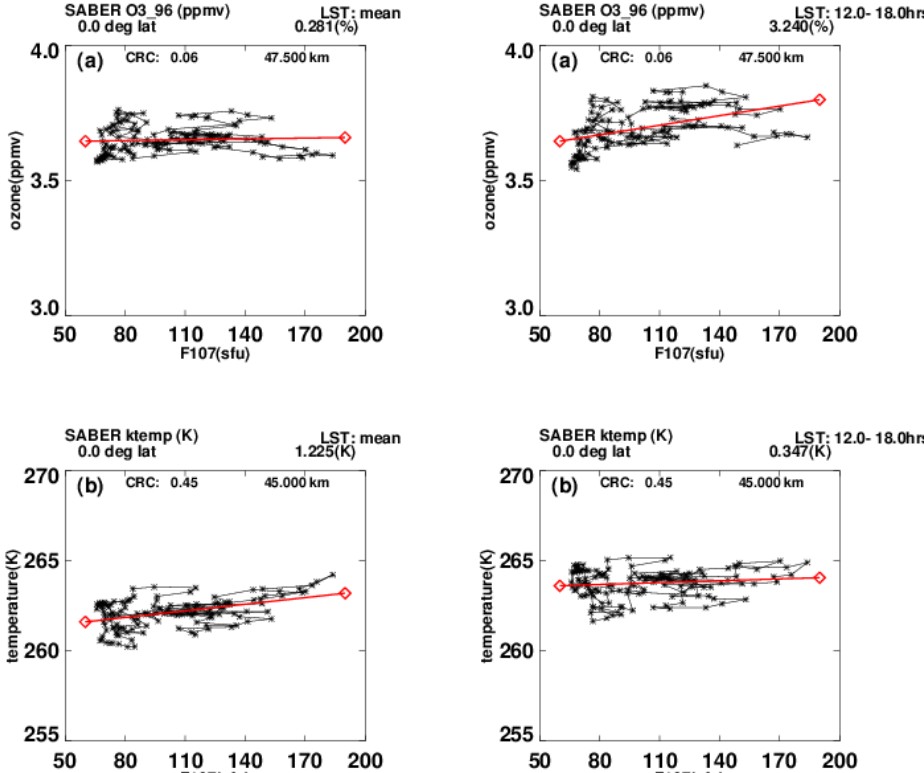

**Figure 11.** Top row: scatter plot of ozone monthly values versus 10.7 cm flux (sfu) at 47.5 km and the Equator. Left
(a): monthly values are zonal means, including average over local time. Right (b): as in (a), but zonal means include
simulated local time variations of orbital drift. Bottom row: as in upper row, but for temperature monthly values. Red
lines: linear fit between monthly values and 10.7 cm flux. Compare with Figure 8.
**7.0 Summary and discussion.**
Using SABER data, we have investigated the effects of ozone and temperature diurnal
variations on their responses to the solar cycle, from 2002 to 2014, and 20 to 100 km.
We find that for ozone, above ~ 40km, zonal means reflecting specific local times (e.g., 6, 12,
18, 24 hrs) lead to different values of responses compared to each other, and compared to
responses based on zonal means that are averaged over the 24 hours of local time (Figures 6,7).
For temperature, effects of diurnal variations are not negligible at ~30 km and above.
We also have considered the variations of local times themselves due to orbital drifts of
certain operational satellites, and their effects on responses to the solar cycle (Figure 8). The
differences can be significant above ~35 km.
The quality and validity of our analysis are shown in comparisons with responses found by
Beig et al., [2012], and Fadnavis and Beig, [2006], based on HALOE data, which made
measurements only at sunrise and sunset. Comparisons with our corresponding results, based on
SABER measurements, are favorable, both at sunrise and sunset separately, and combined. Our
analysis is robust in that the average of responses at specific local times over a diurnal period of
24 hrs is the same as responses based on zonal means that are averages over longitude and local
time together.



Previous studies based other satellite data generally do not describe their treatment, if any, of
local times, so we cannot compare as for HALOE. Some studies also analyzed data merged from
different sources, with measurements made at different local times. As discussed in Section 5.2
in reference to Figure 9, the results of these studies do not generally agree very well among
themselves.
We do not believe that diurnal variations are the major reason for the discrepancies, as there
are likely other data-related issues. Other reasons for differences may be the conditions and
constraints under which the various measurements were made (see Austin et al., 2008, Crooks
and Gray [2005], Gray et al. [2005], Huang et al. [2016b])
However, diurnal variations should be included as part of the analysis of the differences
among various results.
The effects due to satellite orbital drift (discussion in reference to Figure 8) may explain some
unexpected variations in the responses, especially above 40 km.
**Data availability**
The SABER data are freely available from the SABER project at http://saber.gats-inc.com/.
**Acknowledgements.**

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
