# Peer review of "Ozone and temperature decadal solar-cycle responses, and their relation to diurnal variations in the stratosphere, mesosphere, and lower thermosphere, based on measurements from SABER on TIMED."

_Annales Geophysicae, 2019_

## Referee Comment (RC1) · Anonymous Referee #1 · 19 Apr 2019

Overall this paper has some intriguing information but it is presented in a confusing way and does not go far enough in showing the reader the changes in diurnal ozone & temperature values on a global scale. This reviewer recommends that the changes measured between solar max and minimum be plotted as a function of latitude. We believe that the diurnal changes are different at different latitudes (fig 6 of Diurnal ozone variations in the stratosphere revealed in observations from the Superconducting Submillimeter-Wave Limb-Emission Sounder (SMILES) on board the International

Space Station (ISS) by Sakazaki et al) and that the maximum diurnal cycle occurs at 60 degrees latitude in the summer months so the question that needs to be addressed is: does the solar cycle affect ozone and temperature differently at different latitudes? If there is no difference in the changes vs latitude, then this needs to be explicitly stated early in this paper. If there is, then plots for zonal averages (10, 20 or even 30 degrees) is necessary. This could be very useful information for the satellite retrieval community as well as fodder for the modelers to compare to. Also, a short discussion of instrument/measurement error bars would be extremely helpful.

Specific comments: Line 30: based on Line 39: The understanding of the response. . .. Line 154: responses due to the solar. . ...

Figure 1 is extremely jumbled- please remove all trailing zeros (unless you know your altitude registration to 1 meter. . ..:ˆ) what does "data 2005001 2005365" mean on the plot when the caption says 2005085?

Figure 2: Please explain "znimn" in the figure caption or remove.

Line 250,258: change 20006 to 2006

Line 253-4. "The comparisons will indicate the quality of our results. . ." Does it? Either remove or expand.

Line264-5: As stated in the beginning of this review, if there are latitudinal changes in the diurnal cycle between solar min and max, please show us! This is very useful information. Or are you saying the responses change due to increased noise and shouldn't/can't be shown?? Either way, this reviewer feels that showing two latitude bands on the globe are not enough to make the point.

Line 274; should that be figure 3 (not 4)?

Line 306: where are the uncertainties discussed? Line 307: please discuss your error bars [and/or reference]

Figures 3-8: explain LSTNRM in caption or remove.

Figures 6,7 and 8 contain the interesting results of this paper. Again, a more comprehensive paper showing different latitudes in 10, 20 or 30 degree bands would be useful and enlightening.

Section 5.2 This reviewer can't help but feel that some numbers games are being played here. You compare SABER from 24s to 24n to Bieg 0-30 north and south separately. All the others are 25n to 25s (I believe- what latitudes are the red plusses??) so I recommend just removing the Beig data.

Line 518 Previous studies based on. . ..
* * *

---

## Referee Comment (RC2) · Anonymous Referee #2 · 1 May 2019

The manuscript presents an attempt to estimate interference between the decadal solar cycle and diurnal cycle in temperature and ozone profiles using SABER measurements. This type of study would be useful for the satellite community to reconcile observed differences in the response to the decadal solar cycle associated with the differences in measurement times. However, the manuscript needs a major revision, and in its current state does not provide clear conclusions and evidences. My general comments are provided below. General comments: -There is essentially no description of the SABER dataset used in this study and preliminary steps taking to create zonal means that are analyzed in this study. There is a brief mentioning of interpolation, but it is not clear whether this interpolation is required and how it can alter the final dataset. Authors show that the response on solar cycle can be different at different local times, but it's not clear if these differences are statistically significant and not aliasing from differences in sampling across local times or regression model etc. -The analysis is based on multi-regression model, where some terms could be cross-correlated. There is no discussion whether this model is appropriate for the study, what are the uncertainties of this model, and how these uncertainties can affect the derived results. -In this paper authors mostly focuses on the equatorial region, but they never provided a motivation for doing this. Are responses on the solar cycle larger in the equatorial band? It would be helpful if author can summarize their results and provide a global map identifying altitudes and latitudes where the differences in responses are stronger due to differences in measurement time. -The main motivation of this paper is to demonstrate that the response on the solar decadal cycle could be different depending on solar local time. Authors claim that this effect can explain a large fraction of differences in the solar responses reported in previous studies. I hoped that Section 5 can shed light on this issue and offer some explanation based on results of this study. Instead authors show responses on the solar cycle in O3 and temperature from many different instruments leaving readers to wonder why the results are different and could it be due to differences in measurement time. Specific comments: Line 21: Suggest to replace "Our results of responses" by "Responses derived in this study"; Line 43-44: this statement requires a reference. Also, it might be better to say "the magnitude of responses"; Line 47-49: Currently this statement reads like there were no detailed studies on the diurnal cycle, while there are numerous studies on this topic. I assume you meant that previously nobody considered connections between the diurnal cycle and solar decadal cycle. Line 51: does "global empirical results" refer to responses on the 11-year solar cycle? Then replace it with "...previously global responses on the 11-year Solar cycle from empirical measurements . . ." Lines 78-83: this exact paragraph

is repeated again (lines 400-405). Is there any specific reason for doing this? Line 84: On the first two pages authors many times mentioned "previous results" and that they don't agree with each other. It would be helpful to be more specific and say something like: "In study A the ozone response on the solar cycle at altitude X km was Y DU, while study B claimed only Z DU". Otherwise, these statements look very vague. Line 107-108: Section 4 shows results for a few local times, not for all 24-hours. Section 2.0. Some basic information regarding to SABER measurements should be provided here: altitude range, vertical resolution, space and temporal sampling. Figure 1 and the corresponding legend: On all plots it says that results are shown in Line 188: What does it mean "consistent with 3D models"? Line 189-190: This statement is confusing. Do you mean "...our earlier results"? Figure 2: what is the purpose of Figure 2? Since this paper is about responses on the solar cycle at different local times, I have difficulty to understand why the ozone time series are shown here considering its 0.06 correlation with the solar cycle. Line235-238: Please, state how did you define solar maximum and minimum. Is that a month where the F107 flux has it's minimum/maximum, or an average over a few months around that time? Line 253: replace "8" with "18"; Line 255-256 and Sec. 3.1: Is there better way to show HALOE results rather than "manually transferred values". Can you reach out to authors of the study and ask for the dataset? Also, this section list so many reasons why HALOE and SABER results might differ that by the end of this section I fill that there is no value in comparing them. Figure 3, caption: replace "solar activity" with "solar decadal cycle". Figure 4, caption, line 316: It should be first explained that these are results based on HALOE analysis and then the reference should be given. Section 4: it would be useful to show the response on the solar 11-year cycle as a function of solar local time for several altitude levels (similar to fig. 1). Line 391-394: it is not clear from the context what "global results" are refer to. Is it global response on the solar decadal cycle? Section 5: I am not sure what is the purpose of this section. Authors heavily criticized previously published studies because the diurnal effect wasn't taking into account. In this section, results from previous studies are collected, but authors do not offer any explanation for the

observed spread in the results. Does diurnal effect explain the differences? Line 476: should be "at the Equator" Section 6 and Figure 11: The figure has two a) panels and two b) panels, and I was not able to understand what is shown on those plots. Reading section 6 didn't help me to understand that either. This section and figure should be revised.
* * *

---

## Author Comment (AC1) · 12 May 2019

**1a) Referee #1:** Overall this paper has some intriguing information but it is presented in a confusing way and does not go far enough in showing the reader the changes in diurnal ozone & temperature values on a global scale. This reviewer recommends that the changes measured between solar max and minimum be plotted as a function of latitude. We believe that the diurnal changes are different at different latitudes (fig 6 of Diurnal ozone variations in the stratosphere revealed in observations from the Superconducting Submillimeter-Wave Limb-Emission Sounder (SMILES) on board the International Space Station (ISS) by Sakazaki et al) and that the maximum diurnal cycle occurs at 60 degrees latitude in the summer months so the question that needs to be addressed is: does the solar cycle affect ozone and temperature differently at different latitudes?

   **Response 1a):** Before responding to specifics, we wish to note the intended length and scope of the manuscript.
   As it stands, at different latitudes, the variation of the responses to the decadal solar cycle can be seen in Figure 3(4ºlat), Figure 5 (32º, 16º), Figure 6 (16º), and Figure 7 (Equator).
   **In response to the reviewer for more figures, we added an Appendix with 4 plots/2 figures, corresponding to Figure 7 of the manuscript, but at 32ºN and 44ºN latitude.**
   **Also in response to the reviewer, we have added errors bars to Figures 6, 7, 8, and to the added Figures A1 and A2 in the Appendix. However, we did not add error bars to other figures, as they seem to only make the plots busier, and sometimes can make the details more difficult to discern. Besides, the errors are quite consistent from figure to figure because the SABER data are extremely stable, with few dropouts.**

   **The revised and new figures are included below at the end of this response.**

As for adding even more figures, the manuscript is already long, more than 20 pages, and adding more of what the reviewer suggests would be well outside the scope.

To explain why the manuscript is already long, we note the following:

1) Unlike previous results, there is the added variable of local time in addition to latitude and altitude.

2) In addition to the extra variable of local time, there have been essentially no previous studies on the effects of diurnal variations, over the 24 hrs of local time, on the responses of ozone and temperature to the decadal solar cycle (~11 years),. Because nearly all relevant results are new, and we need to spend space to substantiate the validation and reality of the results.

3) We derive responses to the solar cycle for
   a) both ozone and temperature
   b) in the stratosphere, mesosphere, and lower thermosphere,

Usually, previous results by others in this area (even without regard to diurnal variations), cover the stratosphere and mesosphere in separate papers, and often ozone and temperature in separate papers.

For example, we compare various results with results based on HALOE data with Beig et al., [2012] and Fadnavis and Beig [2006], who separated their studies into two papers.

In addition to latitude, our higher priorities are also the variations of the responses to the solar cycle as a function of altitude, because the diurnal variations of ozone and temperature themselves are relative small in the stratosphere, and can dominate in the upper mesosphere and lower thermosphere. As expected, the effects due to diurnal variations on the responses can be large at high altitudes. What was unexpected, at least to us, was that the diurnal effects were not negligible even at low altitudes in the stratosphere.

The point here is that much of the results and discussion can only be basic, limited by space and scope.

Concerning the diurnal variations themselves, we agree that the diurnal variations themselves are a function of latitude, as shown by our previous papers (e.g., Huang et al, 2010b), in addition to the results by Sakazaki et al.,[2013]. We have added the Sakazaki et al., [2013] reference to the manuscript.

In item 11) below, Referee#1 states "… a more comprehensive paper showing different latitudes in 10, 20 or 30 degree bands would be useful and enlightening.

We agree.

This is our point as well, and we could readily write a more comprehensive paper, concentrating on details and variations with latitude. However, that should be for another day.

**1b) Referee #1:** If there is no difference in the changes vs latitude, then this needs to be explicitly stated early in this paper. If there is, then plots for zonal averages (10, 20 or even 30 degrees) is necessary. This could be very useful information for the satellite retrieval community as well as fodder for the modelers to compare to. Also, a short discussion of instrument/measurement error bars would be extremely helpful.

**Response1b):**
As stated earlier, the variation with latitude can be seen in Figures 3(4°lat), 5 (32°, 16°), 6(16°), and 7 (Equator).

Also as stated earlier, what we have done in response to the reviewer is to add an Appendix with 4 plots/2 figures corresponding to Figure 7 of the manuscript, for 32ºN and 44ºN latitude.

Also in response to the reviewer, we have added errors bars to Figures 6, 7, 8, and to the new Figures A1 and A2 in the Appendix of the manuscript. However, we did not add error bars to other figures, as they seem to only make the plots busier, and sometimes can make the details more difficult to discern. The errors are quite consistent from figure to figure because the SABER data are extremely stable, with few dropouts.

**We have added a Section 2.2.2 (Statistical and error considerations) to the manuscript to describe our treatment of uncertainties, as follows:**

**"2.2.2 Statistical and error considerations**
The analysis of uncertainties is the same for the current study as the previous study of the mean variations just described. It is only the input data that are different. Previously, the input consisted of zonal means that are averaged over both longitude and local time, as in 3D models. Here the zonal mean reflect measurements made at specific local times. Details of the statistical analysis are given in Huang et al.,[2106a, 2016b].
The studies use a least squares fit of the multiple regression of Equation (1). Uncertainties in the responses are found from the sample variance (Bevington and Robinson, 1992, Huang et al., 2016a) of the fit. The curvature matrix and its inversion are quite stable due to the excellent sampling of SABER, as there are essentially no significant data dropouts to speak of. So the standard errors are quite stable and reasonable, as can be seen in the error bars in Figures 6, 7, 8, and A1 and A2, in the Appendix. Although very stable in our case, the inversion of the curvature matrix does not explicitly or definitively address potential aliasing among the various terms of the multiple regression, unless the matrix is diagonal.
In Section 6 (Data length and aliasing) below, we show that the derived responses are essentially the same whether we use all the terms in Equation (1) or only the term containing the solar flux. So aliasing is not an issue here."

**Specific comments:**

**2) Referee #1:** Line 30: based on Line 39: The understanding of the response: : :.
Line 154: responses due to the solar: : :..

**Response 2):** Done. We thank the referee for noticing.

**3) Referee #1:** Figure 1 is extremely jumbled- please remove all trailing zeros (unless you know your altitude registration to 1 meter: : :.ˆ) what does "data 2005001 2005365" mean on the plot when the caption says 2005085?

**Response 3):** We have revised the figure according to the reviewer.
The extra information was for 'bookkeeping" purposes only, and has been removed.

**4) Referee #1:** Figure 2: Please explain "znimn" in the figure caption or remove.

**Response 4):** "znlmn" denotes zonal mean

**5) Referee #1:**Line 250,258: change 20006 to 2006
    **Response 5):** Done. We thank the referee for noticing.

   **6) Referee#1:**Line 253-4. "The comparisons will indicate the quality of our results: : :" Does it? Either remove or expand.

**Response  6):** In relevant parts of the manuscript, we have given our opinion about the quality of results in comparisons with results by Beig et al., [2012] and Fadnavis and Beig [20006], based on HALOE data. Although we believe that the comparisons are good, they are by necessity subjective, because the HALOE results are given in 30º latitude composites. As discussed in the manuscript, according to the authors, the sampling of the HALOE data is routinely sparse, and responses are estimated using data over a 30º latitude bin. They do not describe exactly how the data are composited, but in any case, we cannot duplicate it. We get results at 4º degree latitude intervals, so quantitative comparisons should not be made.

**7) Referee #1:** Line264-5: As stated in the beginning of this review, if there are latitudinal changes in the diurnal cycle between solar min and max, please show us! This is very useful information. Or are you saying the responses change due to increased noise and shouldn't/can't be shown?? Either way, this reviewer feels that showing two latitude bands on the globe are not enough to make the point.

   **Response 7):** We are perplexed. Nowhere (lines 264-265 or otherwise) do we even mention 'increased noise and shouldn't/can't be shown' concerning our data. Perhaps the reviewer is reading into what we state about the HALOE data, as opposed to our results.
   As mentioned in response 6) above, for comparison with HALOE, we state that according to the authors, uncertainties in the HALOE data need to be considered, the main problem being routine sparse data. Consequently, HALOE responses are presented in composite 30º latitude bins. The authors do not describe exactly how they treat the data in order to derive responses, but they would not be averages over individual latitudes.
   We get results at 4º latitude-intervals, and from everything that we have seen, there are no problems. In comparing with HALOE we would not be comparing exactly the same things, even if we averaged.  So we are not sure what the reviewer means about 'noise and shouldn't be shown.'
   Again, our comparisons with HALOE are necessary qualitative, but we believe are at least good.
   We agree that showing our results at only two latitudes does not describe global variations as a function of latitude adequately.
   But the fact that they are different at the two latitudes does show that there are variations with latitude.
   In any case, we have added in the Appendix, 4 plots/Figures A1 and A2, depicting results at 32º and 44º.  We have also added error bars to these plots, as well as to Figures, 6,7, and 8.
   Again, in 11) below, Referee#1 states "… a more comprehensive paper showing different latitudes in 10, 20 or 30 degree bands would be useful and enlightening.

This is our point as well, and we could readily write a more comprehensive paper, concentrating on details and latitude.  However, that should be for another day.

**8) Referee#1:** Line 274; should that be figure 3 (not 4)?

   **Response 8):** We did mean Figure 4, and we realize that the sentence is confusing at that point. We have removed the sentence because Figure 4 is discussed in more details in the paragraph after the next.

**9) Referee#1:**  Line 306: where are the uncertainties discussed? Line 307: please discuss your error bars [and/or reference]
   **Response 9):** As stated in our response 1b), above, we have added errors bars to Figures 6,7, 8, A1, A2 of the manuscript. However, we do not think it useful to add error bars to other figures, as they seem to only make the plots busier. The errors are quite consistent from figure to figure because the SABER data are extremely stable, with few dropouts.
   As stated earlier, we have added Section 2.2.2 (Statistical and error considerations) to the manuscript to describe our treatment of uncertainties.
   It is given in quotes in the response to 1b). Also, aliasing among various terms in the regression are minimal. These are all supported by the discussion in Section 6 (Time span of measurements) of the manuscript, where it is found that the derived responses are essentially the same whether we use the all the terms in Equation (1) or only the term containing the solar flux.

**10) Referee#1:**Figures 3-8: explain LSTNRM in caption or remove.

   **Response 10):** As noted in the manuscript, the ozone responses are presents in percent. The normalization depends on the situation. When comparing with HALOE, the normalization would be ozone values at sunrise/sunset. When comparing with zonal means that are averaged over local time, as in Figures 6 and 7, the normalization would also be average over local time.

**11) Referee#1:** Figures 6,7 and 8 contain the interesting results of this paper. Again, a more comprehensive paper showing different latitudes in 10, 20 or 30 degree bands would be useful and enlightening.

   **Response  11):**  As stated earlier, we have added in the Appendix Figures A1 and A2, depicting results at 32º and 44º. As noted in responses 1a), 1b), we are already covering the stratosphere, mesosphere, and lower thermosphere, for both ozone and temperature. We are not aware of any other study that has covered this much. We agree with the reviewer that a more comprehensive paper would be helpful.

**12) Referee#1:** Section 5.2 This reviewer can't help but feel that some numbers games are being played here. You compare SABER from 24s to 24n to Bieg 0-30 north and south separately. All the others are 25n to 25s (I believe- what latitudes are the red plusses??) so I recommend just removing the Beig data.

**Response 12):** We take exception to the reviewer's remarks about 'numbers games'. As a matter of principle, we avoid such games.

We included Figure 9 in the manuscript because readers might ask why, besides HALOE, we did not compare results with other previous studies. Figure 9 was taken intact from a previous paper by us [Huang et al. 2016b], to described previous results by others, based on a variety of data. As noted in the manuscript, these previous results did not describe how they address diurnal variations. The effects of diurnal variations on the responses were not a consideration for them. So comparisons would not be fruitful.

To answer the reviewer's question, in the current manuscript, in discussing Figure 9, we noted that "The red line (plusses) in Figure 9(a) show ozone responses from Soukharev and Hood [2006] (AUDTA, data from1979-2003), as reported by Austin et al. [2008], and from models (AUMDL, magenta lines and triangles), also reported by Austin et al. [2008], representing composite results from 25ºS to 25ºN latitude. The Soukharev and Hood [2006] results (red plusses) are a composite based on SBUV, HALOE, and SAGE data, …"

Note that the red plusses represent results in the latitude interval 25ºS to 25ºN.
That's why our results are averaged over 24ºS to 24ºN (4-degree intervals).

Also note that their analysis used combined SBUV, SAGE, and HALOE data, which mixed measurements at different local times.

Austin et al.,[2012] discussed the differences among the results, and we would agree that they need to be explained. Because of the differences in the other results, we added Beig's results separately, to provide more information conveniently (so long as we made clear that the results were for 30º, we do not believe that it was confusing).

We also did not endeavor to explain the differences, as there are other data-related issues, as noted in the abstract and Summary and discussion section of the manuscript, where we state "We do not believe that diurnal variations are the major reason for the discrepancies, as there are likely other data-related issues. Other reasons for differences may be the conditions and constraints under which the various measurements were made (see Austin et al., 2008, Crooks and Gray [2005], Gray et al. [2005], Huang et al. [2016b])."

**We have added a paragraph to the beginning of Section 5.2, as follows:**

"Unlike the above comparisons with results by Beig et al.,[2012] based on HALOE data, other studies, such as those based on operational satellites, generally did  not describe how the approached the issue of diurnal variations in detail. We will not then attempt to make comparisons, but only present some previous findings. In addition to issues related to local times, there are been reports based on data-related issues in general. Details can be found in Austin et al., [2008], Crooks and Gray [2005], Gray et al. [2005], and Huang et al. [2016b]."

**13) Referee#1:** Line 518 Previous studies based on: : :.
  **Response 13):** We thank the reviewer for noticing.

[Figure]

**Figure 6.** Ozone (left panel) and temperature (right) responses from 50 to 100 km at 16ºN. Values are responses at solar max minus responses at solar min (% /100sfu) for ozone and ºK/100sfu for temperature. Black asterisks denote responses based on zonal means that are averages over both longitude and local time. Green asterisks denote our responses based on zonal means fixed at 6hrs, blue diamonds fixed at 12hrs, red plusses at 18 hrs, and magenta triangles at 24hr, based on SABER data.

[Figure]

**Figure 7.** As in Figure 6, but from 20 to 60 km. Ozone (left panel) and temperature (right) responses at 0º. Values are responses at solar max minus responses at solar min (% /100sfu) for ozone and ºK/100sfu for temperature. Black asterisks denote our responses based on zonal means that are averages over both longitude and local time. Green asterisks denote our responses of zonal means at 6hrs, blue diamonds at 12hrs, red plusses at 18 hrs, and magenta triangles at 24hrs, based on SABER data.

[Figure]

**Figure 8.** Ozone (left panel) and temperature (right panel) responses to solar activity versus altitude, at the Equator, from 20 to 60 km. Values are responses at solar max minus responses at solar min in % per 100 sfu for ozone, and K/100 sfu for temperature. Black asterisks denote responses based on zonal means that are averages over both longitude and local time. Red squares denote corresponding results, but with local times increasing linearly from 12 to 18 hrs from 2002 to 2014.

[Figure]

**Figure A1.** As in Figure 7, Ozone responses at 32° (left panel) and 44° from 20 to 60 km. Values are responses at solar max minus responses at solar min (% /100sfu) . Black asterisks denote our responses based on zonal means that are averages over both longitude and local time. Green asterisks denote our responses of zonal means at 6hrs, blue diamonds at 12hrs, red plusses at 18 hrs, and magenta triangles at 24hrs, based on SABER data.

[Figure]

**Figure A2.** As in Figure A1. temperature responses at 32º (left panel) and 44º, from 20 to 60 km. Values are responses at solar max minus responses at solar min (ºK/100sfu). Black asterisks denote our responses based on zonal means that are averages over both longitude and local time. Green asterisks denote our responses of zonal means at 6hrs, blue diamonds at 12hrs, red plusses at 18 hrs, and magenta triangles at 24hrs, based on SABER data.

**References:**

Bevington, P. R. and Robinson, D. K.,: Data reduction and error analysis for the physical sciences, McGraw-Hill, New York, USA, 1992.

Huang, F. T., Mayr, H. G., Russell III, J. M., and Mlynczak, M.G.: Ozone and temperature decadal responses to solar variability in the mesosphere and lower thermosphere, based on measurements from SABER on TIMED, Ann. Geophys., 34, 29–40, doi:10.5194/angeo-34-29-2016, 2016a.

Huang, F. T., H. G. Mayr, J. M. Russell III, and M. G. Mlynczak, Ozone and temperature 601 decadal responses to solar variability in the stratosphere and lower mesosphere, based on 602 measurements from SABER on TIMED, Ann. Geophys., 34, 801–813, doi:10.5194/angeo-34-603 801-2016, 2016b.

Soukharev, B. E., and L. L. Hood (2006), The solar cycle variation of stratospheric ozone: Multiple regression analysis of long-term satellite data sets and comparisons with models, J. Geophys. Res., 111, D20314, doi:10.1029/2006JD007107.

---

## Author Comment (AC2) · 12 May 2019

**I) Reviewer#2:** The manuscript presents an attempt to estimate interference between the decadal solar cycle and diurnal cycle in temperature and ozone profiles using SABER measurements. This type of study would be useful for the satellite community to reconcile observed differences in the response to the decadal solar cycle associated with the differences in measurement times. However, the manuscript needs a major revision, and in its current state does not provide clear conclusions and evidences. My general comments are provided below.

  **Response I):** Before responding to specifics, we wish to note the length and scope of this manuscript, regarding additional figures.
   **In response to the reviewer for more figures, we added an Appendix with 4 plots/2 figures corresponding to Figure 7 of the manuscript, but at 32ºN and 44ºN latitude.**
   **Also in response to the reviewer, we have added errors bars to Figures 6, 7, 8, and to the added Figures A1 and A2 in the Appendix. However, we do not add error bars to other figures, as they seem to only make the plots busier, and sometimes can make the details more difficult to discern. In addition, the errors are quite consistent from figure to figure because the SABER data are extremely stable, with few dropouts.**

   **The revised and new figures are included below at the end of this response.**

   As for adding more figures, the manuscript is already long, more than 20 pages, and adding more of what the reviewer suggests would be well outside the scope.
   To explain why the manuscript is already long, and adding figure would expand the manuscript too much, note the following:
   1) Unlike previous results, there is the added variable of local time in addition to latitude and altitude.
   2) In addition to the extra variable of local time, there have been essentially no previous studies on the effects of diurnal variations, over the 24 hrs of local time, on the responses of

ozone and temperature to the decadal solar cycle (~11 years),. So nearly all relevant results are essentially new, and we need to spend space to substantiate the validation and reality of the results.

> 3)  We derive responses to the solar cycle for
> > a) both ozone and temperature
> > b) in the stratosphere, mesosphere, and lower thermosphere,

Because of the wide ranges that are covered, our results can only be basic in nature.

Usually, previous results by others in this area (even without regard to diurnal variations), cover the stratosphere and mesosphere in separate papers, and often ozone and temperature in separate papers.

For example, we compare various results with results based on HALOE data with Beig et al., [2012] and Fadnavis and Beig [2006], who separated their studies into two papers.

The point here is that much of the results and discussion can only be basic, limited by space and scope.

**A) Reviewer#2: General comments:**

**A0) Reviewer#2:** -There is essentially no description of the SABER dataset used in this study and preliminary steps taking to create zonal means that are analyzed in this study. There is a brief mentioning of interpolation, but it is not clear whether this interpolation is required and how it can alter the final dataset.

**Response A0): We have updated the heading to Section 2.0 and added the following:**

**"2.0 SABER data characteristics and analysis.**
The SABER/TIMED instrument [Russell et al., 1999] was launched in December 2001 with an orbital inclination of~74º. SABER views the Earth's limb to the side of the orbital plane, and vertical profiles, corresponding to the line-of-sight tangent point, are retrieved from measurements of the $CO_2$ 15 and 4.3 μm emissions for kinetic temperature, and from the 9.6μm channel for ozone. About every 60 days, TIMED is yawed by 180º, so that the SABER measurement footprint of SABER is ~83ºN-52ºS or 83ºS to 52ºN on alternate yaw periods. Over a given day and for a given latitude circle, measurements are made as the satellite travels northward (ascending mode) and again as the satellite travels south-ward (descending mode). Data at different longitudes are sampled over 1 day as the Earth rotates relative to the orbit plane. SABER scans altitude (~10-105 km for temperature, 15-100 km for ozone) every 58s with an altitude resolution of ~2km, with ~96 scans per orbit, and ~14 longitudes per day.
The orbital characteristics of the satellite are such that, over a given day, a given latitude circle, and a given orbital mode (ascending or descending), the local time at which the data are measured is essentially the same, independent of longitude and time of day. For a given day, latitude, and altitude, we work with data averaged over longitude: one for the ascending orbital mode and one for the descending mode, each corresponding to a different local solar time, resulting in two data points for each day. Each can be biased by the local time variations and is therefore not a true zonal mean. True zonal means are averages made at a specific time over longitude around a latitude circle, with the local solar time varying by 24 h over 360° in longitude. The local times of the SABER measurements decrease by about 12 min from day to day, and it takes 60 days to sample over the 24 hrs of local time."

Regarding interpolation, as with most data sets, measurements are not made at regular latitude or altitude grids. Common methods for gridding include interpolation or binning. We interpolate to 4º latitudes and 2.5 km altitude based on the sampling of SABER.

We have also tested binning for previous papers (diurnal variations) and found that the results are virtually the same. In Figure 10 of the manuscript, we compare our results with those of Nath and Sridharan (2014), who analyzed the same SABER data as did we, and who (presumably) binned the data in the 10-15º latitude band. As can be seen our results, from data interpolated to 12º, are very similar for altitudes below 45km, where diurnal variations for both ozone and temperature are relatively small. As noted in the manuscript, it does not appear than Nath and Sridharan (2014) considered effects of local time variations, which would explain the more obvious differences above 45 km.

Regardless, the agreement below 45 km shows that binning and interpolation provides very similar results, considering the difference in the treatment of diurnal variations.

**A1) Reviewer#2:** Authors show that the response on solar cycle can be different at different local times, but it's not clear if these differences are statistically significant and not aliasing from differences in sampling across local times or regression model etc.
-The analysis is based on multi-regression model, where some terms could be crosscorrelated.

**Response A1):** In previous papers, we had discussed uncertainties in the results (responses not involving diurnal variations) using the same algorithm (see our answer to A2) below, including possible aliasing in the multiple regression. We should not assume that referencing them alone would be adequate.

**Therefore, to the manuscript, we have added a section (2.2.2 Statistical and error considerations), as follows:**

**"2.2.2 Statistical and error considerations**
The analysis of uncertainties is the same for the current study as the previous study of the mean variations just described. It is only the input data that are different. Previously, the input consisted of zonal means that are averaged over both longitude and local time, as in 3D models. Here the zonal mean reflect measurements made at specific local times. Details of the statistical analysis are given in Huang et al.,[2106a, 2016b].
The studies use a least squares fit of the multiple regression of Equation (1). Uncertainties in the responses are found from the sample variance (Bevington and Robinson, 1992, Huang et al., 2016a) of the fit. The curvature matrix and its inversion are quite stable due to the excellent sampling of SABER, as there are essentially no significant data dropouts to speak of. So the standard errors are quite stable and reasonable, as can be seen in the error bars in Figures 6, 7, 8, and A1 and A2, in the Appendix. Although very stable in our case, the inversion of the curvature matrix does not explicitly or definitively address potential aliasing among the various terms of the multiple regression, unless the matrix is diagonal.
In Section 6 (Data length and aliasing) below, we show that the derived responses are essentially the same whether we use all the terms in Equation (1) or only the term containing the solar flux. So aliasing is not an issue here."

We have added error bars to Figures 6, 7, 8, and to new Figures A1 and A2, in the Appendix.

For more on aliasing and cross correlation in the multiple regression, we refer the reviewer to Section 6 and Figure 11 of the manuscript. We recognize the reviewer has explicit questions about this as well in (B20), below.

**We have updated the heading of Section 6.0 to 'Data length and aliasing', and added to the discussion of Figure 11 to increase clarity, as follows:**
" In Section 2.2.2, we noted that in the application of Equation (1), possible aliasing among the different terms are not definitively addressed. In addition, it has been argued that more than one solar cycle is needed. Following our analysis given in Huang et al.,[2016b], we address these issues in this section. "

**A2) Reviewer#2:** There is no discussion whether this model is appropriate for the study, what are the uncertainties of this model, and how these uncertainties can affect the derived results.

**Response A2):** We assume that by 'model', the reviewer refers to the multiple regression, Equation (1). In Section 2.2, in discussing the multiple regression, we state "The estimates of responses to the solar cycle are made using Equation (1), in a similar manner as previously done by others, and by us, using a multiple regression analysis (e.g., Keckut et al. [2005], Soukharev and Hood [2006],…" The multiple regression had been previously used by numerous authors, although we explicitly referenced only two. We add that almost all papers in this area use the same basic multiple regression as we do, and as we have in Huang et al.,[2016a, 2016b].
Since it has been used so often in the past, we guess that the reviewer is asking about how this fits in with diurnal variations, which previous studies have not considered.
The connection is in the input M(t) in Equation (1). For diurnal variations, we generate the ozone or temperature zonal means at the desired local time for input to Equation (1). We repeat for other local times as needed. It is similar to previous studies using data from HALOE or from sun-synchronous satellites, which measures at one or two local times only.
Since we can generate M(t) at any day and local time for input, we can then generate responses to the solar cycle for any given local time.
This is how we can compare with HALOE explicitly, at 6 and 18 hrs.
The regression equation is

$$M(t) = a + b*t + d*F107(t) + c*S(t) + l*lst(t) + g*QBO(t) \qquad (1)$$

where t is time (months), a is a constant, b is the trend , $d$ the coefficient for solar activity (10.7 cm flux), c is the coefficient for the seasonal *(S(t))* variations, $l$ the coefficient for local time *(lst)* variations, and $g$ the coefficient for the QBO. As is often done, the seasonal and local time variations are removed first, but we include them in Equation (1) for completeness. The F107 stands for the solar 10.7 cm flux, which is commonly used as a measure of solar activity, and the values used here are monthly means provided by NOAA.
M(t) stands for the input ozone or temperature zonal means, either at specific local times (current application), or averaged over local times (previous studies).

**Uncertainties:** The derivation of uncertainties are addressed in our response to A1) above. As stated in response to (A1) we have added error bars to Figures 6, 7, 8, and to new Figures A1 and A2, in the Appendix.

**A3) Reviewer#2:** -In this paper authors mostly focuses on the equatorial region, but they never provided a motivation for doing this. Are responses on the solar cycle larger in the equatorial band? It would be helpful if author can summarize their results and provide a global map identifying altitudes and latitudes where the differences in responses are stronger due to differences in measurement time.

**Response A3):** There is no physical reason that we start with equatorial regions. We began with the equatorial region to compare with previous results, both with and without effects of diurnal variations. Examples are Austin et al.,[2008], and Beig et al.,[2012], who presented results in the equatorial region, as in Figures 3, 4, and 9 of the manuscript. Our higher priorities are also the variations of the responses to the solar cycle as a function of altitude, because the diurnal variations themselves of ozone and temperature themselves are relatively small in the stratosphere, and can dominate in the upper mesosphere and lower thermosphere. As expected, the effects due to diurnal variations on the responses can be large at high altitudes. What was unexpected, at least to us, was that the diurnal effects were not negligible even at low altitudes in the stratosphere.

In addition, we tried to substantiate our results (and those of HALOE as well) in comparison with Beig et al., [2012], and Fadnavis and Beig [2006], at sunrise and sunset.

We did this for both ozone and temperature.

The point here is the constraint of space and length of the manuscript.

As stated in the manuscript, we have results from 20 to 100km and 48ºS to 48ºN latitude. We also have results for both ozone and temperature. Because there have been essentially no previous comparable results, we needed to also consider the reality of our results, and we compared results with that based on HALOE data at some length.

We have not considered more latitudes because we just have too many results and need to be selective.

We have added Figures A1 and A2 in the Appendix, corresponding to Figure 7 of the manuscript, showing responses of ozone and temperature at 32º and 44º.

In our paper Huang et al.,[2016b], where we looked at responses, but averaged over diurnal effects, we did provide in Figure 5 of that paper global contours. To provide similar contours over 24 hrs of local time would take up much more space.

Our main goals are to examine if diurnal variations do affect the responses to solar cycles, and if so, to examine to their basic extent. We do this for both ozone and temperature, in the stratosphere, mesosphere, and lower thermosphere.

Further details are beyond the scope of the manuscript.

The other anonymous reviewer also wanted more results at more latitudes. However, he/she did volunteer that would be for another manuscript.

We would agree. We could readily generate a separate manuscript with the added information, but that is for another day,

The manuscript is already well over 20 pages.

**A4) Reviewer#2:** -The main motivation of this paper is to demonstrate that the response on the solar decadal cycle could be different depending on solar local time. Authors claim that this

effect can explain a large fraction of differences in the solar responses reported in previous studies.

**Response A4):** We agree with the reviewer that "The main motivation of this paper is to demonstrate that the response on the solar decadal cycle could be different depending on solar local time." However, we do not think that we have claimed that diurnal effects can "explain a large fraction of differences reported in… previous studies."

In the introduction (lines 85-88) and Summary and discussion (lines 562-565), we state

"We do not believe that diurnal variations are the major reason for the discrepancies, as there are likely other data-related issues. Other reasons for differences may be the conditions and constraints under which the various measurements were made (see Austin et al., 2008, Crooks and Gray [2005], Gray et al. [2005], Huang et al. [2016b])."

However, diurnal variations should be included as part of the analysis of the differences among various results."

We also state "The effects due to satellite orbital drift (discussion in reference to Figure 8) may explain some unexpected variations in the responses, especially above 40 km."

**A5) Reviewer#2:** I hoped that Section 5 can shed light on this issue and offer some explanation based on results of this study. Instead authors show responses on the solar cycle in O3 and temperature from many different instruments leaving readers to wonder why the results are different and could it be due to differences in measurement time.

**Response A5):** We understand the reviewer's disappointment, and wish that the agreements among other were better. We felt that we had to mention other results besides those from HALOE since readers may ask about them. As noted above, in the introduction (lines 85-88) and Summary and discussion (lines 562-565), we state

"We do not believe that diurnal variations are the major reason for the discrepancies, as there are likely other data-related issues. Other reasons for differences may be the conditions and constraints under which the various measurements were made (see Austin et al., 2008, Crooks and Gray [2005], Gray et al. [2005], Huang et al. [2016b])."

Although we give references, perhaps we should have emphasized "other data-related issues" more.

In our paper Huang et al., [2016b] we stated "As noted by Crooks and Gray (2005), "In summary, [. . . ] results support the growing body of evidence that variability associated with the 11-year solar cycle has a significant influence on stratospheric temperatures. However, there is still no consensus on the exact magnitude and spatial structure; longer and more consistent satellite observations are needed to resolve this issue.""

We also stated that "In comments about the inconsistencies of the various studies, Crooks and Gray (2005) also state:" "We note here that tests have shown that none of the discrepancies between the current work and that of S2000 and H2004 can be explained simply in terms of the slightly different lengths of the various datasets employed, nor the fact that H2004 used the Mg II index to represent solar variability rather than the 10.7-cm radio flux as was used in the current study and in S2000. We suggest that differences between the datasets employed is the primary reason for the large disagreement between the results of H2004 and those shown in the current analysis and in S2000.""

Austin et al., [2008] describe some details of discrepancies among the various results and 3D models.

In Section 5.0 of the manuscript, we noted that, unlike Beig et al.,[2012], the various studies generally did not address the issue of diurnal variations in detail. Consequently, it is not possible to try and separate effects of diurnal variations from 'other data related issues' in these various studies.

But we remind the reviewer that we have accomplished the following:

a) that diurnal variations do have significant and systematic effects on the response of ozone and temperature to the solar cycle.

b) that the effects in the upper mesosphere and lower thermosphere are large, as perhaps can be expected, since the diurnal variations of ozone and temperature themselves can be dominant in the higher altitudes.

c) that even in the stratosphere, the effects of diurnal variations on the responses can still be significant, even though the diurnal variations of ozone and temperature themselves are relatively small in the lower altitudes.

d) changes in the local times due to orbital drift over years can have systematic effects on the derived responses, especially above 40 km.

We had 'known', even before this study, that there were probable issues with much of the data used previously.

**B) Specific comments:**

**B1) Reviewer#2:** Line 21: Suggest to replace "Our results of responses" by "Responses derived in this study";
  **Response B1):** Done.

**B2) Reviewer#2:** Line 43-44: this statement requires a reference. Also, it might be better to say "the magnitude of responses";
  **Response B2):** Done.

**B3) Reviewer#2:** Line 47-49: Currently this statement reads like there were no detailed studies on the diurnal cycle, while there are numerous studies on this topic. I assume you meant that previously nobody considered connections between the diurnal cycle and solar decadal cycle.
  **Response B3):** We have made the sentence clearer

**B4) Reviewer#2:** Line 51: does "global empirical results" refer to responses on the 11-year solar cycle? Then replace it with "...previously global responses on the 11-year Solar cycle from empirical measurements : : :"
  **Response B4):** Done, except 'empirical measurements' is redundant.

**B5) Reviewer#2:** Lines 78-83: this exact paragraph is repeated again (lines 400-405). Is there any specific reason for doing this?
  **Response B5):** We wanted to reiterate this relevant issue in the manuscript.
We have reworded and deleted some phrases, so they are not exact.

**B6) Reviewer#2:** Line 84: On the first two pages authors many times mentioned "previous results" and that they don't agree with each other. It would be helpful to be more specific and say something like: "In study A the ozone response on the solar cycle at altitude X km was Y DU, while study B claimed only Z DU". Otherwise, these statements look very vague.

   **Response B6):** The whole paragraph stated, "Previous results have not generally agreed so well with one another in their details. A major reason for these differences may be the conditions and constraints under which the various measurements were made (Austin et al., 2008, Crooks and Gray [2005], Gray et al. [2005], Huang et al. [2016b])." We have pointed the reader specifically to the references, especially Austin et., [2012], who describe the differences in some detail. Also in the Summary and discussion.

**B7) Reviewer#2:** Line 107-108: Section 4 shows results for a few local times, not for all 24-hours.

   **Response B7): We have added the following paragraph to the beginning of Section 4:** "Although the figures show responses only at 6, 12, 18, and 24 hrs, we have generated hourly responses, and can do so at any local time. We do not believe that plots at additional local times would add important information for purposes here, and would make other details less discernible."

**B8) Reviewer#2:** Section 2.0.
Some basic information regarding to SABER measurements should be provided here: altitude range, vertical resolution, space and temporal sampling.

   **Response B8):** See our response to (A0), above. We have added to Section 2.0 of the manuscript.

**B9) Reviewer#2:** Figure 1 and the corresponding legend: On all plots it says that results are shown in Line 188: What does it mean "consistent with 3D models"?

   **Response B9):** The zonal means of 3D models are averages over both longitude and local time. The zonal means based on data that are measured only at one fixed local time reflect averages only over longitude. The local time is fixed at the value where the measurement is taken. This point was also made by Austin et al.,[2008].

**B10) Reviewer#2:** Line 189-190: This statement is confusing. Do you mean ": : :our earlier results"?

   **Response B10):** Yes. We have added 'earlier'.

**B11) Reviewer#2:** Figure 2: what is the purpose of Figure 2? Since this paper is about responses on the solar cycle at different local times, I have difficulty to understand why the ozone time series are shown here considering its 0.06 correlation with the solar cycle.

**Response B11):** As explained in the text, the green lines show how the data would behave if the local times of the measurements changed due to orbital drift. It merely gives the reader a better qualitative view of what can be expected. Although this description is in the text, we neglected to describe the green line in the figure caption. It has been added.

**B12) Reviewer#2:** Line235-238: Please, state how did you define solar maximum and minimum. Is that a month where the F107 flux has it's minimum/maximum, or an average over a few months around that time?

**Response B12):** Solar max is the month where the 10.7 cm flux is max, solar min is the month where the f10.7 is min.
Shown in Figure 2.

**B13) Reviewer#2:** Line 253: replace "8" with "18";
**Response B13):** Done. We thank the reviewer for noticing.

**B14) Reviewer#2:** Line 255-256 and Sec. 3.1: Is there better way to show HALOE results rather than "manually transferred values". Can you reach out to authors of the study and ask for the dataset? Also, this section list so many reasons why HALOE and SABER results might differ that by the end of this section I fill that there is no value in comparing them.
**Response B14):** We have not asked the authors for their numbers. Their papers are many years old, and we feel confident that our transcription is accurate. We have been careful to print their figures and used rulers to measure the numbers. Most importantly, in comparing the plots visually, we could not discern differences.
We mentioned this only to be professional and transparent.

**B15) Reviewer#2:** Figure 3, caption: replace "solar activity" with "solar decadal cycle".
**Response B15):** Done

**B16) Reviewer#2:** Figure 4, caption, line 316: It should be first explained that these are results based on HALOE analysis and then the reference should be given. Section 4: it would be useful to show the response on the solar 11-year cycle as a function of solar local time for several altitude levels (similar to fig. 1).

**Response B16):** We have inserted the reference to HALOE.
Although we appreciate the reviewer's interest, we think that this would open up a new line of inquiry and should be part of anther manuscript.
We have explained in the beginning of this response why the manuscript is already long
Some of the information that the reviewer wants can be seen in Figures 6 and 7, although only at 4 local times. We have added 2 more figures in the Appendix similar to Figure 7, but at 32° and 44° latitude.

**B17) Reviewer#2:** Line 391-394: it is not clear from the context what "global results" are refer to. Is it global response on the solar decadal cycle?

**Response B17):** We have added 'solar decadal cycle'.

**B18) Reviewer#2:** Section 5: I am not sure what is the purpose of this section. Authors heavily criticized previously published studies because the diurnal effect wasn't taking into account. In this section, results from previous studies are collected, but authors do not offer any explanation for the observed spread in the results. Does diurnal effect explain the differences?

**Response B18):** We do not believe that we criticized, much less heavily criticized, previously published studies. At least that was not our intention. We mentioned it because we could not compare without information on how they handled diurnal variations. If they did, we might have been able to adapt, as we did with HALOE.
   We refer the reviewer to our response to response A5) above for discussion and explanation of differences.

**B19) Reviewer#2:** Line 476: should be "at the Equator"
   **Response B19):** Done, although we think that 'at' also works.

**B20) Reviewer#2:** Section 6 and Figure 11: The figure has two a) panels and two b) panels, and I was not able to understand what is shown on those plots. Reading section 6 didn't help me to understand that either. This section and figure should be revised.

**Response B20):** Section 6 and Figure 11 address directly the reviewer's comments in A1) and A2) above, concerning crosscorreltion (aliasing as used by us) and also comments about the length of the data.
**We have changed the heading to Section 6.0 and added the following:**

   " In Section 2.2.2, we noted that in the application of Equation (1), possible aliasing among the different terms are not definitively addressed. In addition, it has been argued that more than one solar cycle is needed. Following our analysis given in Huang et al.,[2016b], we address these issues in this section. "
   We refer the reviewer to our response A1) and A2), above.

Figures

[Figure]

**Figure 6.** Ozone (left panel) and temperature (right) responses from 50 to 100 km at 16ºN. Values are responses at solar max minus responses at solar min (% /100sfu) for ozone and ºK/100sfu for temperature. Black asterisks denote responses based on zonal means that are averages over both longitude and local time. Green asterisks denote our responses based on zonal means fixed at 6hrs, blue diamonds fixed at 12hrs, red plusses at 18 hrs, and magenta triangles at 24hr, based on SABER data.

[Figure]

**Figure 7.** As in Figure 6, but from 20 to 60 km. Ozone (left panel) and temperature (right) responses at 0º. Values are responses at solar max minus responses at solar min (% /100sfu) for ozone and ºK/100sfu for temperature. Black asterisks denote our responses based on zonal means that are averages over both longitude and local time. Green asterisks denote our responses of zonal means at 6hrs, blue diamonds at 12hrs, red plusses at 18 hrs, and magenta triangles at 24hrs, based on SABER data.

[Figure]

**Figure 8.** Ozone (left panel) and temperature (right panel) responses to solar activity versus altitude, at the Equator, from 20 to 60 km. Values are responses at solar max minus responses at solar min in % per 100 sfu for ozone, and K/100 sfu for temperature. Black asterisks denote responses based on zonal means that are averages over both longitude and local time. Red squares denote corresponding results, but with local times increasing linearly from 12 to 18 hrs from 2002 to 2014.

[Figure]

**Figure A1.** As in Figure 7, Ozone responses at 32° (left panel) and 44° from 20 to 60 km. Values are responses at solar max minus responses at solar min (% /100sfu) . Black asterisks denote our responses based on zonal means that are averages over both longitude and local time. Green asterisks denote our responses of zonal means at 6hrs, blue diamonds at 12hrs, red plusses at 18 hrs, and magenta triangles at 24hrs, based on SABER data.

[Figure]

**Figure A2.** As in Figure A1. temperature responses at 32° (left panel) and 44°, from 20 to 60 km. Values are responses at solar max minus responses at solar min (°K/100sfu). Black asterisks denote our responses based on zonal means that are averages over both longitude and local time. Green asterisks denote our responses of zonal means at 6hrs, blue diamonds at 12hrs, red plusses at 18 hrs, and magenta triangles at 24hrs, based on SABER data.

---

## Author Response (AR2)

**Responses to (final) comments of referees 1 and 2 of**

**"Ozone and temperature decadal solar-cycle responses, and their relation
to diurnal variations in the stratosphere, mesosphere, and lower thermosphere, based on
measurements from SABER on TIMED" by Frank T. Huang and Hans Mayr**

**A) Anonymous Referee #1:**
 **There were no additional comments by referee#1**

**B) Anonymous Referee #2:**
 **B1) Anonymous Referee #2:**
Except of one comment I have only very minor technical comment. Therefore I recommend
(very) minor revision.
Page 15: What is the out result from Fig. ? There is only description of Fig. 9 in the paper.

 **Authors' response to B1):** Figure 9 was taken from a previous paper of ours that showed
results (responses to solar cycle) by others using various data, but which ignored effects of local
time/diurnal variations. For convenience, we only wanted to give the reader a quick description
of previous studies that did not consider effects of diurnal variations.

 **B2) Anonymous Referee #2:**
Line 30: 'based other' should be based on other'
Line 170: delete one dot at the end of the sentence
Line 198: "Sakazaki et al.' – add year
Line 229: 'solar the cycle' should be 'the solar cycle'
Line 363 delete 'Left Panel:'
Line 462: delete comma – 'al.,[' should be "al['
Line 487: 'there are been' should be 'there have been'

List or references:
-References Barasseur (1193), Haigh et al. (2004), Maycock et al. (2016), Mitchell et al. (2014)
And Shindell et al. (1999) are in the List ofreferences but I did not find a reference to them in the
body of the paper. Either refer to them in the paper, or delete them.

-Line717: Move Remsberg et al. (2002) to a separate line.

 **Authors' response to B2:**
We have corrected all of the above issues noted by reviewer #2.
We thank the reviewer for noticing.

June 3May 14, 2019

[revised manuscript text omitted]